# Effectiveness of Lower-Cost Strategies for Running Gait Retraining: A Systematic Review

**Lissandro M. Dorst** [1] , **Vitor Cimonetti** [1], **Jefferson R. Cardoso** [2], **Felipe A. Moura** [3] and **Rodrigo R. Bini** [4,*]

1. Laboratory of Applied Biomechanics, State University of Londrina, Londrina 86057-970, Brazil
2. Laboratory of Biomechanics and Clinical Epidemiology, PAIFIT Research Group, State University of Londrina, Londrina 86057-970, Brazil
3. Laboratory of Applied Biomechanics, Sport Sciences Department, State University of Londrina, Londrina 86057-970, Brazil
4. Rural Health School, La Trobe University, Bendigo, VIC 3550, Australia
* Correspondence: r.bini@latrobe.edu.au

**Abstract:** The effectiveness of lower-cost equipment used for running gait retraining is still unclear. The objective of this systematic review was to evaluate the effectiveness of lower-cost equipment used in running gait retraining in altering biomechanical outcomes that may be associated with injuries. The literature search included all documents from MEDLINE, Web of Science, CINAHL, SPORTDiscus, and Scopus. The studies were assessed for risk of bias using an evaluation tool for cross-sectional studies. After screening 2167 initial articles, full-text screening was performed in 42 studies, and 22 were included in the systematic review. Strong evidence suggested that metronomes, smartwatches, and digital cameras are effective in running gait retraining programs as tools for intervention and/or evaluation of results when altering step cadence and foot strike patterns. Strong evidence was found on the effectiveness of accelerometers in interventions with feedback to reduce the peak positive acceleration (PPA) of the lower leg and/or footwear while running. Finally, we found a lack of studies that exclusively used lower-cost equipment to perform the intervention/assessment of running retraining.

**Keywords:** biomechanics; runners; feedback; cadence; foot strike pattern

## 1. Introduction

The number of people who run has increased in recent years due to the associated health benefits, accessibility, and low cost, making it one of the most popular forms of exercise [1]. However, due to the repetitive nature, overuse injuries are common. Studies suggest that between 42.7% and 79% of runners experience an injury in any given year [2–4].

The etiology of running injuries is known to be multifactorial in nature and includes intrinsic (advanced age, higher body mass index, history of previous injury, discrepancy in limb length, abnormal anatomical alignment and foot posture, altered foot load patterns, individual abilities, and cognitive properties) and extrinsic (ground surface, footwear, and training load) factors [5–14]. Common running injuries include plantar fasciopathy, Achilles tendinopathy, medial tibial stress syndrome, patellofemoral pain, iliotibial band syndrome, patellar tendinopathy, tibial stress fracture, hamstring injury including proximal tendinopathy, and gluteal tendinopathy [2,15,16].

Several strategies for the prevention and treatment of running injuries are applied by coaches and runners themselves; these include stretching, warming up, technical training, and changing the running technique (called retraining) to reduce the load on certain muscle groups and joints [17]. Biomechanical studies have extensively examined running retraining strategies that include changes in the step cadence, stride length, distance between the heel and the center of mass at the initial foot contact with the ground, duration of flight phase, foot strike pattern, hip and knee movement, trunk position, step width, and impact load

variables, among others; these studies also reported changes in the variables of kinematics, kinetics, and electromyography [18]. In this sense, several studies have documented the alterations in running mechanics in runners who develop injuries such as excessive pronation of the foot [19,20], accentuated hip adduction [21], increased internal hip rotation, contralateral pelvic drop, and a reduction in the peak hip flexion, among others [22].

In recent years, studies have proposed changes to running techniques (i.e., movement) through running retraining with feedback in order to reduce impact loads (force applied to the skeleton when contact with the ground occurs) as a way to reduce injury risk [23–32]. Outcome measures concentrated on determining the effectiveness of running retraining using visual and/or auditory feedback in real time to modify kinematics and kinetics during running [22,33–36]. The step rate was shown to significantly reduce impact forces in long-distance runners with an increase of only 5% in step cadence [24]. The study identified that by increasing the step rate, kinematic variables such as the step length, vertical oscillation of the center of mass, and angle of inclination of the foot were reduced. The reductions in these variables were associated with decreased impact forces that could theoretically reduce the risk of injuries. Other authors supported a reduction in impact forces by increasing the cadence or rate of steps [37–39]. Increasing step cadence with a proportional reduction in the stride length at a constant speed has been shown to facilitate a reduction in foot inclination angles and impact forces. This decreases the number of initial contacts on the ground by the hindfoot during the step [37,39,40]. In this sense, some studies analyzed the change in the strike pattern and proposed a shift from stepping on the rearfoot to stepping on the middle or forefoot because the impact forces on the knees and hips are typically higher for heel striking compared to midfoot or forefoot striking [37,41,42]. It was reported that approximately 80% of recreational runners who use traditional running shoes opted for heel striking [43,44]. Thus, changing the strike pattern through gait retraining could be a way to reduce impact forces and the risk of injuries related to running [35,45–48].

A systematic review with meta-analyses to assess the effectiveness of running retraining on kinematics, kinetics, performance, pain, and injury in long-distance runners [49] found that gait retraining was effective in increasing the step cadence and reducing the mean vertical load rate. It was also observed that gait retraining to minimize heel striking increased the knee flexion at the initial contact. However, trials that reported on peak tibial acceleration (in the skin surrounding the tibia) and the peak patellofemoral joint reaction force were too different to pool their data. Results from individual trials demonstrated reductions in these outcomes across multiple retraining interventions. However, in a recent systematic review of randomized clinical trials on strategies to prevent and manage running-related knee injuries [50], low-quality evidence was found to indicate that retraining on running techniques may reduce the risk of running-related knee injuries by two-thirds. These findings highlighted the effectiveness of gait retraining in runners to alter movement-related risk factors that are potentially associated with the development of musculoskeletal injuries.

Even though the efficiency of running retraining in reducing the risk of injuries is debatable, many studies and clinicians still utilize these interventions to improve outcomes for individual runners. However, most studies seem to rely on high-cost instruments such as force platforms (on the ground or on treadmills) and three-dimensional (3D) motion-analysis systems to provide real-time biofeedback. Unfortunately, these devices are unaffordable to many and are rarely available to coaches or clinicians. Clinical equipment generally includes a simple treadmill, a high-definition video camera, and computer applications or smartphones to identify variables such as the step cadence, foot strike angle, and foot strike pattern [40,51].

There is still a need to examine the effectiveness of lower-cost equipment utilized in running retraining interventions. Therefore, the objective of this systematic review was to evaluate the effects of lower-cost equipment on running gait retraining. As a definition, in

this study we assumed that lower-cost equipment would have a reduced cost compared to gold-standard devices.

## 2. Materials and Methods

### 2.1. Databases and Search Strategy

This systematic review was carried out in accordance with the principles of the Preferred Reporting Items for Systematic Reviews and Meta-Analyses (PRISMA) and registered on the Open Science Framework (OSF) website (https://osf.io/z4uxm). Searches were performed in the MEDLINE, Web of Science, CINAHL, SPORTDiscus, and Scopus databases; the searches were limited to publications in English, excluded reviews and congress abstracts, and had no date restrictions. Articles were searched using the following search strategy (identical for all databases) from the first year of database registration until September 2022: ((((((("running") OR "jogging") OR "run") OR "track and field") OR "runners")) AND ((((("retraining") OR "retrain") OR "feedback") OR "biofeedback")) AND (((((((((("injury") OR "Injuries") OR "injured") OR "lesion") OR "disability") OR "contusion") OR "disease") OR "disorder") OR "pain")).

### 2.2. Selection of Studies

The studies were selected by two reviewers (L.M.D. and V.C.), and a third reviewer (R.R.B.) was available to resolve any disagreements regarding the final eligibility of selected publications. All studies identified by the search strategy were exported to EndNote version X8 (Clarivate Analytics) by one investigator. First, the removal of duplicate articles was performed automatically. Next, an analysis of the titles of all identified studies was performed by the reviewers followed by abstracts and full text.

Studies were accepted or excluded based on inclusion and exclusion criteria. To be included, studies needed to: (1) involve interventions using running retraining with feedback (no time limitation); (2) use lower-cost equipment as a tool for intervention and/or evaluation of the intervention results; (3) report biomechanical variables; and (4) use predictors of risk of injury or pain attenuation [18]. Studies were excluded if they reported interventions that did not include running retraining, running without feedback, participants with prosthetic limbs, neurological or congenital impairments, the use of only expensive equipment in the study, or children or participants under 18 years of age.

### 2.3. Data Extraction and Analysis

The data extracted from each article included: authorship, year of publication, sample characteristics, division of sample groups, participant demographics, intervention protocols, method of providing feedback, equipment used, analyzed variables related to lower-cost equipment, and the results of these variables. The data analysis was based on running parameters modified through retraining when lower-cost equipment was used.

### 2.4. Risk of Bias Analysis

Eligible studies were assessed in terms of their risk of bias using an assessment tool for cross-sectional studies (the AXIS tool). This assessment tool, which was developed by Downes et al. [52], aims to assist in the interpretation of a study and inform decisions about the quality thereof. The AXIS tool consists of 20 components to examine study quality, study design, and the potential risk of bias in cross-sectional studies [52]. Each question can be answered as yes, no, unable to determine, or not applicable; the scoring consists of one point for "yes" and zero points for "no", unable to determine", or "not applicable". Some criteria were excluded from the analysis because they were not related to the evaluated studies (criteria 7, 13, and 14); therefore, 17 criteria contributed to the final score. The number of "yes" responses was calculated to determine the percentage of the criteria that were met in each study.

*2.5. Data Synthesis for Evidence-Based Recommendations*

Outcomes were synthesized for each study using a modified model of the van Tulder criteria [53]:

- Strong evidence: findings were consistent across at least three studies, two of which were of high quality.
- Moderate evidence: findings were consistent across at least two studies, one of which was of high quality.
- Limited evidence: findings were consistent across one high-quality study or two low- or moderate-quality studies.
- Very limited evidence: findings were consistent across a moderate or low-quality study.
- Inconsistent evidence: results were inconsistent across multiple studies.
- Conflicting evidence: results were contradictory across multiple studies.
- No evidence: findings were negligible regardless of study quality.

### 3. Results

The literature search identified 2167 articles in the five databases searched. Of these, 2125 studies were removed after screening for duplicates and reading of titles and abstracts, which resulted in 42 studies that were independently read by two reviewers. After reading the full-text articles, 12 studies were excluded due to not using equipment considered to be lower-cost (e.g., three-dimensional motion-analysis systems and force plate), 2 studies for presenting case studies, 1 for lack of focus on feedback as an intervention, 1 because it provided data on the same sample as in another original study, and 4 articles for not analyzing outcomes for the lower-cost equipment used. Thus, a total of 22 studies were included for the analysis of this systematic review. Figure 1 presents the study-selection flowchart.

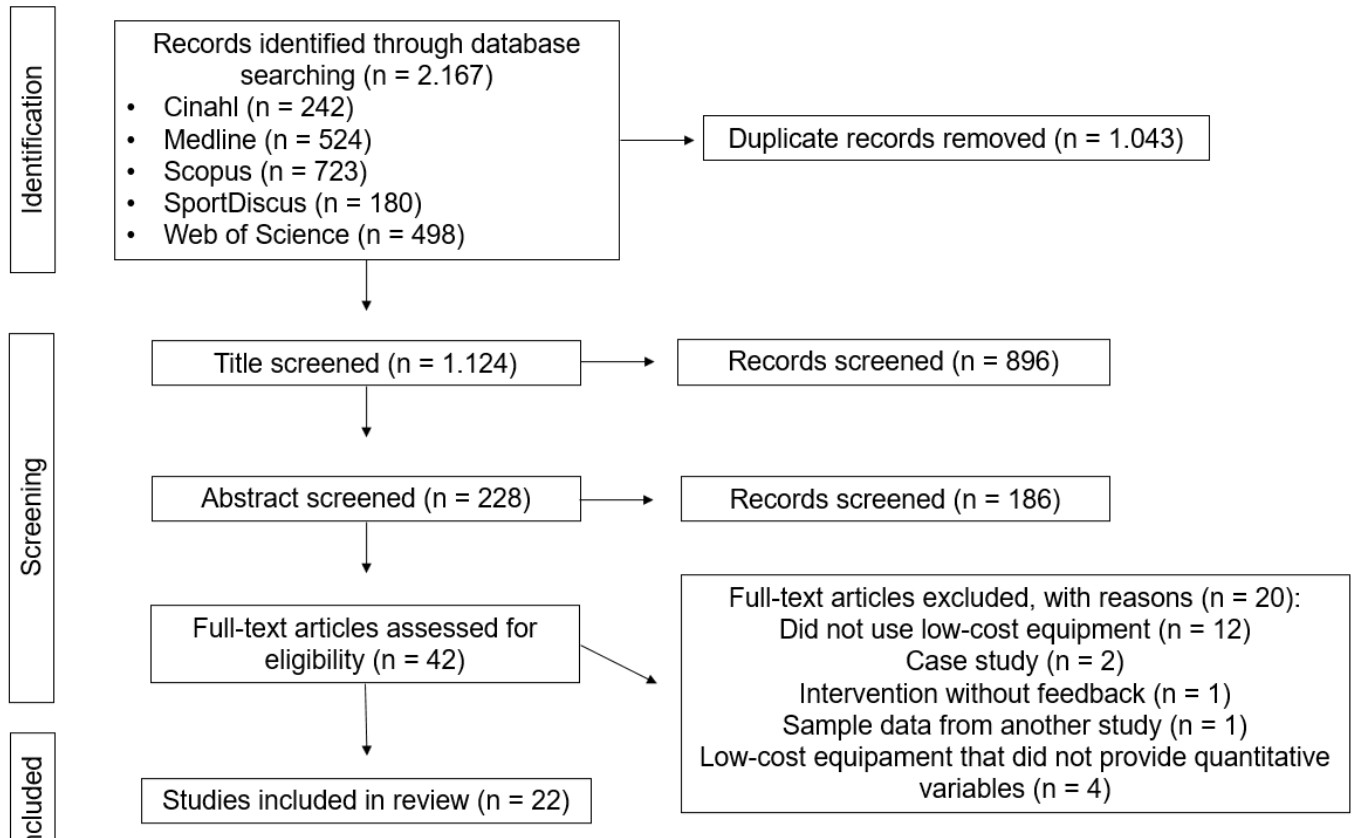

**Figure 1.** Preferred Reporting Items for Systematic Reviews and Meta-Analyses (PRISMA) flowchart of the included studies.

### 3.1. Risk-of-Bias Assessment of Included Studies

The risk of bias of all eligible studies was assessed using the AXIS tool [52]. The assessment of quality and risk of bias indicated that all studies were of a very high quality, with 54% of studies meeting 82% of the criteria, 32% meeting 88% of the criteria, and 14% meeting 94% or more of the criteria (Table 1). Among the main limitations of the studies were a lack of justification regarding the sample size, the non-representativeness of the sample as a target population, and a lack of clarity in terms of recruitment.

### 3.2. Analyzed Studies

An analysis of the lower-cost equipment used in the 22 studies included in this systematic review (Tables 2–5) showed that 45.5% of the studies used equipment such as smartwatches with an accelerometer, metronomes, stopwatches, video cameras, platforms, instrumented socks and insoles with sensors to control cadence and/or the foot strike pattern, and the distribution of plantar pressure and the peak force in contact with the ground [40,47,48,54–60]. We also found that 54.5% of the studies used an accelerometer and/or an inertial central unit to identify the peak positive acceleration (PPA) of the tibia (the skin surrounding the tibia) and/or footwear [30–32,61–69].

The results of studies that aimed to increase the preferred cadence reported effectiveness in modifying the step frequency and presented significant increases after the intervention that varied from 6% to 8.6% [40,48,54,55,58–60]. In addition, studies that used lower-cost equipment to change the foot strike pattern by suggesting that runners not land on their heels also obtained very satisfactory clinical results [47,48,56]. In addition, a study by Goss et al. [47] that used a digital camera and instrumented socks showed that 95% of the runners transitioned to a foot strike pattern other than heel striking after the retraining (Tables 2 and 3).

Studies that analyzed the tibial and/or shoe PPA with the use of accelerometers showed significant reductions in the tibial and/or shoe PPA after performing the retraining [32,61–69]. These reported significant reductions did not present normative values because diverse methodologies of interventions were applied to the runners (Tables 4 and 5).

**Table 1.** Assessment of methodological quality using the AXIS scale. Y = criterion met, N = criterion not met. Final score = sum of Ys and Ns in the case of criterion 19 (with the percentage value in parentheses). Some criteria were excluded from the analysis because they were not related to the studies evaluated (criteria 7, 13, and 14); therefore, 17 criteria contributed to the final score.

| Included Studies | Criteria | | | | | | | | | | | | | | | | | Final Score (%) |
|---|---|---|---|---|---|---|---|---|---|---|---|---|---|---|---|---|---|---|
| | 1 | 2 | 3 | 4 | 5 | 6 | 8 | 9 | 10 | 11 | 12 | 15 | 16 | 17 | 18 | 19 | 20 | |
| Allen et al. (2016) [40] | Y | Y | N | Y | Y | Y | Y | Y | Y | Y | Y | Y | Y | Y | Y | N | Y | 16 (94) |
| Baumgartner et al. (2019) [55] | Y | Y | Y | Y | Y | Y | Y | Y | Y | Y | Y | Y | Y | Y | Y | N | Y | 17 (100) |
| Cheung et al. (2018) [66] | Y | Y | N | Y | N | N | Y | Y | Y | Y | Y | Y | Y | Y | Y | N | Y | 14 (82) |
| Cheung et al. (2019) [31] | Y | Y | Y | Y | N | N | Y | Y | Y | Y | Y | Y | Y | Y | Y | N | Y | 15 (88) |
| Ching et al. (2018) [30] | Y | Y | N | Y | N | N | Y | Y | Y | Y | Y | Y | Y | Y | Y | N | Y | 14 (82) |
| Clansey et al. (2014) [63] | Y | Y | N | Y | N | N | Y | Y | Y | Y | Y | Y | Y | Y | Y | N | Y | 14 (82) |
| Creaby and Smith (2016) [65] | Y | Y | N | Y | N | N | Y | Y | Y | Y | Y | Y | Y | Y | Y | N | Y | 14 (82) |
| Crowell et al. (2010) [61] | Y | Y | N | Y | N | N | Y | Y | Y | Y | Y | Y | Y | Y | Y | N | Y | 14 (82) |
| Crowell and Davis (2011) [62] | Y | Y | Y | Y | N | N | Y | Y | Y | Y | Y | Y | Y | Y | Y | N | Y | 15 (88) |
| Da Silva Neto, Lopes, and Ribeiro, (2021) [57] | Y | Y | Y | Y | N | N | Y | Y | Y | Y | Y | Y | Y | Y | Y | N | Y | 15 (88) |
| Goss et al. (2021) [47] | Y | Y | Y | Y | N | N | Y | Y | Y | Y | Y | Y | Y | Y | Y | N | Y | 15 (88) |
| Letafatkar et al. (2020) [67] | Y | Y | N | Y | N | N | Y | Y | Y | Y | Y | Y | Y | Y | Y | N | Y | 14 (82) |
| Miller et al. (2021) [58] | Y | Y | Y | Y | Y | N | Y | Y | Y | Y | Y | Y | Y | Y | Y | N | Y | 16 (94) |
| Morris et al. (2020) [48] | Y | Y | Y | Y | N | N | Y | Y | Y | Y | Y | N | Y | Y | Y | N | Y | 14 (82) |
| Musgjerd et al. (2021) [59] | Y | Y | N | Y | N | N | Y | Y | Y | Y | Y | Y | Y | Y | Y | N | Y | 14 (82) |
| Phanpho, Rao, and Moffat (2019) [56] | Y | Y | N | Y | N | N | Y | Y | Y | Y | Y | Y | Y | Y | Y | N | Y | 14 (82) |
| Sellés-Pérez et al. (2022) [60] | Y | Y | N | Y | N | N | Y | Y | Y | Y | Y | Y | Y | Y | Y | N | Y | 14 (82) |
| Sheerin et al. (2020) [68] | Y | Y | Y | Y | N | N | Y | Y | Y | Y | Y | Y | Y | Y | Y | N | Y | 15 (88) |
| Van den Berghe et al. (2022) [69] | Y | Y | N | Y | N | N | Y | Y | Y | Y | Y | Y | Y | Y | Y | N | Y | 14 (82) |
| Willy et al. (2016) [54] | Y | Y | Y | Y | N | N | Y | Y | Y | Y | Y | Y | Y | Y | Y | N | Y | 15 (88) |
| Wood and Kipp (2014) [64] | Y | Y | N | Y | N | N | Y | Y | Y | Y | Y | Y | Y | Y | Y | N | Y | 14 (82) |
| Zhang et al. (2019) [32] | Y | Y | Y | Y | N | N | Y | Y | Y | Y | Y | Y | Y | Y | Y | N | Y | 15 (88) |

Score: 1–18 and 20: "yes" (Y) = 1, "no" (N) = 0. 19; "no" = 1, "yes" = 0. Criteria: (1) the study goals/objectives were clear; (2) the study design was appropriate for the stated objectives; (3) the sample size was justified; (4) the target/reference population was clearly defined; (5) the sample framework was taken from an appropriate population base to closely represent the target/reference of the population under investigation; (6) the selection process was likely to select subjects/participants who were representative of the target/reference population under investigation; (8) the risk factor and outcome variables were measures appropriate to the objectives of the study; (9) the risk factor and outcome variables were measured correctly using instruments/measurements that were previously tested or published; (10) it was clear what was used to determine statistical significance (e.g.; *p*-values; CIs); (11) the methods were sufficiently described to allow them to be repeated; (12) the baseline data were adequately described; (15) the results were internally consistent; (16) the results of the analyses were described in the methods presented; (17) the authors' discussions and conclusions were justified by the results; (18) the limitations of the study were discussed; (19) there was no source of funding or conflict of interest that could affect the authors' interpretation of the results; and (20) ethics approval or consent of the participants was obtained.

**Table 2.** Summary of experimental studies that analyzed spatiotemporal variables, kinetics, and foot kinematics as a retraining strategy using a control/comparative group.

| Author (year) | Sample Characteristics | Control/Comparative Group (n) | Intervention | Equipment Used | Variables Analyzed (Lower-Cost Equipment) | Results (Lower-Cost Equipment) |
|---|---|---|---|---|---|---|
| Baumgartner et al. (2019) [55] | 38 healthy runners with a preferred cadence of ≤85 steps/minute and running a minimum of 24 km/week. Mean age of the groups: retraining 37.7 SD 9.8 years and control 39.7 SD 14.8 years. | Retraining (20) and control (18) | Duration: 6 weeks Instructions: 10% increase in cadence Feedback: visual | Lower cost: smartwatch and accelerometer | Step cadence | Only the experimental group presented a significant increase ($p < 0.001$) in step cadence (8.6% increase). |
| Da Silva Neto, Lopes, and Ribeiro (2021) [57] | 24 healthy adults with heel strike step type: mean age 44.0 SD 8.9 years and running 25.8 SD 12.1 km/week in the retraining group; and 44.2 SD 8.1 years and running 26.4 years SD 13.5 km/week in the control group. | Retraining (12) and control (12) | Duration: eight sessions in two weeks Instructions: run more smoothly Feedback: visual | Lower cost: pressure platform and stopwatches | Peak pressure, maximum mean pressure, maximum force, and plantar arch | The retraining group presented a reduction in peak pressure in the medial and lateral region of the hindfoot. Maximum force in the midfoot and medial hindfoot region was reduced pre- and post-training in the retraining group and in relation to the control group. The mean maximum pressure did not change with retraining. The plantar arch during running showed a significant increase after retraining, demonstrating an adjustment in plantar support. |
| Morris et al. (2020) [48] | 114 healthy runners with heel strike step type; mean age of the groups: biofeedback (BFB) 25.7 SD 9.1 years and control group (CON) 27.8 SD 9.6 years; running 22.9 SD 14 km/week in the biofeedback group and 23.7 SD 10.7 km/week in the control. | BFB (55) and CON (59) | Duration: single session; BFB received mobile app alert during training for 1 year; analysis of retention after 6 months and 1 year Instructions: verbal cues, exercises for a different type of step than heel strike, soft stepping, and a cadence of 180 steps/minute; groups progressed to a different step than heel strike for 10% of their weekly mileage; the BFB received an alert when the tibial shock exceeded 6 g (6 months) and if there was a heel strike Feedback: auditory and visual | Lower cost: digital camera, treadmill, and inertial measurement unit | Step cadence and step type | 80% of runners demonstrated a different type of foot strike after the 2 h training session. The percentage of non-heel-strike runners at the 6-month and 1-year follow-up decreased slightly in both groups, but was not significant. Both groups presented significant increases in cadence from baseline to post-training (approximately 6%) and from baseline to follow-up at 6 months (approximately 3.7%) and 1 year (approximately 4.2%). |
| Sellés-Pérez et al. (2022) [60] | 12 healthy runners. Mean age of the groups: retraining 35 SD 5.9 years and control 38 SD 7.3 years; running 27 SD 12.7 km/week in the retraining group and 31 SD 11.9 km/week in the control. | Retraining (7) and control (5) | Duration: 6 weeks. Instructions: 10% increase in cadence Feedback: auditory | Lower cost: mobile device (video capture at 60 Hz) and audio player | Step cadence | Only the retraining group presented a significant increase ($p = 0.004$) in stride cadence (7.3% increase). |

**Table 2.** *Cont.*

| Author (year) | Sample Characteristics | Control/Comparative Group (n) | Intervention | Equipment Used | Variables Analyzed (Lower-Cost Equipment) | Results (Lower-Cost Equipment) |
|---|---|---|---|---|---|---|
| Willy et al. (2016) [54] | 30 healthy, high-impact runners. Mean age of the groups: retraining 20.9 SD 1.3 years and control 20.73 SD 1.2 years; running 22.1 SD 7.5 km/week in the retraining group and 23.2 SD 17.9 km/week in the control. | Retraining (16) and control (14) | Duration: eight sessions and analysis of retention after 1 month Instructions: 7.5% increase in step cadence Feedback: visual | Lower cost: smartwatch and triaxial accelerometer High cost: instrumented treadmill | Step cadence | The retraining group showed a significant increase in cadence in the phase immediately after retraining (8.6%) and after one month (8.5%) in relation to the baseline. This significant increase in cadence also occurred in relation to the control group that had an unchanged cadence ($p < 0.0001$). |

**Table 3.** Summary of experimental studies that analyzed spatiotemporal variables, kinetics, and foot kinematics as a retraining strategy without the use of a control/comparative group.

| Author (Year) | Sample Characteristics | Intervention | Equipment Used | Variables Analyzed (Lower-Cost Equipment) | Results (Lower-Cost Equipment) |
|---|---|---|---|---|---|
| Allen et al. (2016) [40] | 40 healthy runners with heel strike type of foot step; mean age 36 SD 9.1 years; running 40 km/week. | Duration: single session Instructions: run at preferred cadence, +5%, +10%, and +15% Feedback: visual and auditory | Lower cost: metronome, video camera (60 Hz), and treadmill | Step cadence, foot inclination angle at the moment of contact with the ground, and type of step | Significant change in the pattern of running from hindfoot to midfoot or forefoot in cadence conditions of +10% and +15% of the preferred cadence in 17.5% and 30% of subjects, respectively. The mean angle of inclination of the foot at the instant of contact with the ground decreased significantly as the cadence increased ($p < 0.001$). |
| Goss et al. (2021) [47] | 19 runners with heel strike step type; mean age 28.8 SD 12 years with a history of injury/surgery in the previous 12 months (20.1 SD 10.9 weeks) but cleared for running by a doctor. Running 8.8 SD 7 km/week. | Duration: 10 sessions and analysis of retention after 1 month Instructions: try to touch the ground more carefully, do not step on the hindfoot, try to lean forward to step on the forefoot, and cadence of 180 steps/minute Feedback: auditory | Lower cost: digital camera, instrumented socks, and an anklet containing an accelerometer High cost: instrumented treadmill | Step type | 95% made the transition to a type of step other than the heel strike, and the majority (89%) maintained the transition from the type of step on retention. |
| Miller et al. (2021) [58] | 9 runners with heel strike step type; mean age 20.3 SD 2.2 years with a history of musculoskeletal injury in the lower limbs in the previous 12 months; duration of injury symptoms of 192.4 SD 345.5 days; released for running by a doctor. | Duration: 10 weeks with six sessions Instructions: do not step on the hindfoot, try to lean forward, run silently, cadence of 180 steps/minute, take shorter and faster steps Feedback: verbal, visual, and auditory | Lower cost: digital camera and metronome High cost: instrumented treadmill | Step type and cadence | 100% of participants transitioned to a different foot strike type after retraining. There was a significant increase in step cadence after retraining of 6.2%. |

**Table 3.** *Cont.*

| Author (Year) | Sample Characteristics | Intervention | Equipment Used | Variables Analyzed (Lower-Cost Equipment) | Results (Lower-Cost Equipment) |
|---|---|---|---|---|---|
| Musgjerd et al. (2021) [59] | 15 healthy runners; mean age 23.5 years; running 16.5 miles/week. | Duration: two sessions up to 10 days apart. Instructions: in the 1st session, participants ran at the self-selected cadence for 2.4 miles; in the 2nd session, the step cadence was increased by 10% and the baseline pace was maintained. Feedback: auditory and visual | Lower cost: instrumented insoles with sensors, smartwatch, and metronome | Step cadence and peak force | There was a significant increase in stride cadence between sessions of 7.3% and a decrease in peak strength of 5.6%. |
| Phanpho, Rao, and Moffat (2019) [56] | 15 healthy runners with heel strike step type; mean age 25.67 SD 3.99 years; ran at least twice a week for at least 30 min. | Duration: single session. Instructions: run with cadence increased by 10% and perform midfoot or forefoot steps. Feedback: visual, auditory, and combined | Lower cost: insoles and socks instrumented with sensors, device built with pedals and microcontroller, and metronome | Location of the center of pressure in relation to the insole | The mean location of the center of pressure at initial contact differed significantly in the feedback conditions in relation to baseline (192.7%) and new cadence (128.5%). However, there was no difference in location between the types of feedback. |

**Table 4.** Summary of experimental studies that analyzed the peak positive acceleration of tibia and footwear as a retraining strategy using a control/comparative group.

| Author (Year) | Sample Characteristics | Control/Comparative Group | Intervention | Equipment Used | Variables Analyzed (Lower-Cost Equipment) | Results (Lower Cost Equipment) |
|---|---|---|---|---|---|---|
| Cheung et al. (2019) [31] | 38 healthy adults (24 for walking retraining and 14 for running retraining); age 26.2 SD 11.2 years; running >12 km/week. | Walking (24) and running (14) | Duration: eight sessions in two weeks. Instructions: run more smoothly and 20% below the footwear PPA baseline average. Feedback: visual | Lower cost: biaxial accelerometer High cost: instrumented treadmill | Footwear PPA and tibial PPA | After retraining, the running group showed a reduction in footwear PPA (40.9%) and tibial PPA (25.8%). Footwear PPA presented values four times higher than the tibial PPA for walking and running. |
| Clansey et al. (2014) [63] | 22 healthy runners with heel strike step type and tibial PPA >9 g. Mean age of the groups: retraining 33.3 SD 9.0 years and control 33.9 SD 11.3 years; running 30.4 SD 7.5 km/week in the retraining group and 35.7 SD 14.2 km/week in the control. | Retraining (12) and control (10) | Duration: six sessions over three weeks and analysis of retention after 1 month. Instructions: information when PPA was above 75%, between 75% and 50%, or below 50% of baseline. Feedback: auditory and visual | Lower cost: triaxial accelerometer High cost: motion capture system, force platform, and photo cells | Tibial PPA | The retraining group showed significant reductions in PPA after training (30.7%) compared with no change in the control group. These modifications were maintained one month after training. |
| Creaby and Smith (2016) [65] | 22 healthy runners. Mean age of the groups: specialist 32.81 SD 7.8 years and accelerometer 22.7 SD 7.8 years; running >10 km/week. | Specialist feedback (GFE) (11) and tibial acceleration feedback (GFAT) (11). | Duration: single session and retention after 7 or 8 days. Instructions: the specialist-provided feedback group was instructed to run more smoothly and with less step noise; the accelerometer feedback group was instructed to run with tibial PPA below 50% of baseline. Feedback: verbal and visual | Lower cost: triaxial accelerometer | Tibial PPA | There was a significant reduction in tibial PPA when compared to baseline in the running with feedback (GFE = 23.9% and GFAT = 28.5%), the running with feedback removed (GFE = 28.1% and GFAT = 18.9%), and retention (GFE = 22.0% and GFAT = 21.2%). No significant differences were found between groups. |

**Table 4.** *Cont.*

| Author (Year) | Sample Characteristics | Control/Comparative Group | Intervention | Equipment Used | Variables Analyzed (Lower-Cost Equipment) | Results (Lower Cost Equipment) |
|---|---|---|---|---|---|---|
| Letafatkar et al. (2020) [67] | 49 healthy adults; the conditioning training (CT) group had a mean age of 33.4 SD 6.25 years, the CT group with feedback had a mean age of 31.2 SD 5.11 years, and the control group had a mean age of 34.2 SD 6.64 years; running >8 km/week for more than 2 years. | CT (16), CT with feedback (17), and control (16) | Duration: 24 sessions in 8 weeks and analysis of retention after 1 year. Instructions: run more smoothly, avoid stepping on the hindfoot, run with the knees apart, and point the patella forward Feedback: verbal and visual | Lower cost: accelerometer High cost: force platform and motion capture system | Tibial PPA | The CT with feedback group presented significant improvement for the tibial PPA after 8 weeks at 8 km/h (38.3%) and at 12 km/h (40.3%) and also in relation to the CT group at 8 km/h, but there was no significant difference at 12 km/h. There was a significant difference for tibial PPA at the 1-year follow-up in the CT group with feedback for 8 and 12 km/h (15.5% and 10.9%) |
| Van den Berghe et al. (2022) [69] | 20 healthy adults with high tibial acceleration; mean age 32.1 SD 7.8 years, PPA of 10.9 SD 2.8 g, and running 27 SD 10 km/week in the experimental group; and 39.1 SD 10.4 years, PPA of 13.0 SD 3.9 g, and running 36 SD 18 km/week in the control group. | Retraining (10) and control (10) | Duration: six sessions over three weeks Instructions: music distortion was related to the PPA and the music was clear when the PPA was 30% below the baseline; when the running speed was changed, a verbal warning was given Feedback: auditory and verbal | Lower cost: accelerometer | Tibial PPA and step cadence | The retraining group presented a significantly decreased PPA (by 25.5%) after retraining without changing cadence. The control group presented no significant change in PPA. |

**Table 5.** Summary of experimental studies that analyzed the peak positive acceleration of tibia and footwear as a retraining strategy without the use of a control/comparative group.

| Author (Year) | Sample Characteristics | Intervention | Equipment Used | Variables Analyzed (Lower-Cost Equipment) | Results (Lower-Cost Equipment) |
|---|---|---|---|---|---|
| Cheung et al. (2018) [66] | 16 healthy runners with footwear PPA >10 g, age 28.3 SD 6.2 years, and running at least 15 km/week. | Duration: eight sessions in two weeks Instructions: running lightly touching the ground during distraction and 20% below the baseline mean of footwear PPA Feedback: visual. | Lower cost: triaxial accelerometer High cost: instrumented treadmill | Footwear PPA | With retraining, PPA showed a significant reduction in the conditions without (41.1%) and with (32.2%) visual feedback and also a significant reduction with visual feedback before (25.7%) and after (14.7%) visual retraining. |
| Ching et al. (2018) [30] | 16 healthy runners with footwear PPA >8 g, age 25.1 SD 7.9 years, and running 16.0 SD 1.7 km/week. | Duration: eight sessions in two weeks Instructions: run with softer steps to avoid the high-pitched sound that was emitted at 80% of the footwear PPA Feedback: auditory. | Lower cost: triaxial accelerometer High cost: instrumented treadmill | Footwear PPA and tibial PPA | There was a reduction in footwear PPA without (33.8%) and with (21.4%) auditory feedback and tibial PPA without (21.5%) and with (20.2%) auditory feedback after retraining. The group exhibited lower footwear and tibial PPA with auditory feedback (22.2% and 9.9%) only before retraining. |

**Table 5.** *Cont.*

| Author (Year) | Sample Characteristics | Intervention | Equipment Used | Variables Analyzed (Lower-Cost Equipment) | Results (Lower-Cost Equipment) |
|---|---|---|---|---|---|
| Crowell et al. (2010) [61] | 5 healthy runners with a mean age of 26 SD 2 years and running a minimum of 32 km/week. | Duration: single session Instructions: maintain the PPA below 50% of the mean Feedback: visual | Lower cost: uniaxial accelerometer High cost: instrumented treadmill | Tibial PPA | 4 out of 5 subjects presented significant reductions in PPA at the end of the no-feedback period compared to the warm-up. The differences between subjects were: −60%, −54%, −36%, −17%, and +6%. |
| Crowell and Davis (2011) [62] | 10 healthy runners with heel strike step type and tibial PPA >8 g with a mean age of 26 SD 7 years and running more than 16 km/week. | Duration: eight sessions in two weeks and analysis of retention after 1 month. Instructions: run smoother, make your steps quieter, and keep your PPA below 50% of the mean Feedback: visual | Lower cost: triaxial accelerometer High cost: force platform | Tibial PPA | PPA was reduced after retraining by 48% and was maintained at one-month follow-up. |
| Sheerin et al. (2020) [68] | 18 healthy runners with high tibial acceleration, mean age of 35.2 SD 9.6 years, and running 42.4 SD 22.2 km/week. | Duration: eight sessions in three weeks and analysis of retention after 1 month. Instructions: run with smoother steps and eliminate vibration feedback, the threshold for which was 10% below the resulting tibial acceleration from the baseline Feedback: tactile | Lower cost: inertial measurement unit, accelerometer, and smartwatch High cost: instrumented treadmill | Tibial PPA | The median of the resulting tibial acceleration pre- and post-intervention in the treadmill running decreased by 50%; while in the ground running, it decreased by 28%. When compared to running on a treadmill, before the intervention and 1 month after the intervention, the median decreased by 41%; while in the ground running, the median decreased by 17%. |
| Wood and Kipp (2014) [64] | 9 healthy runners with heel strike step type, age 20 SD 1.5 years, and running at least 16 km/week. | Duration: single session performed twice: 5 min with biofeedback followed by 5 min without biofeedback Instructions: run with no audio signal from the PPA with a threshold 10 to 15% below the baseline PPA Feedback: auditory | Lower cost: triaxial accelerometer | Tibial PPA | In the 1st period of 5 min of biofeedback, the PPA was significantly reduced (10.2%); and in the 1st period without biofeedback, the PPA did not differ from the baseline. In the 2nd round of biofeedback, the runners significantly reduced their PPA (11.9%); and in the 2nd period without biofeedback, they significantly reduced their PPA from baseline (8.5%). |
| Zhang et al. (2019) [32] | 13 healthy runners with mean tibial PPA >8 g, age 41.1 SD 6.9 years, and running 30.7 SD 22.2 km/week. | Duration: eight sessions in two weeks Instructions: land more softly to avoid reaching 80% of the mean peak of the baseline tibial PPA Feedback: visual | Lower cost: triaxial accelerometer High cost: instrumented treadmill | Tibial PPA | After retraining, PPA was significantly reduced in the trained (35% to 37%) and untrained (22% to 30%) limbs when running at evaluated speeds. |

## 4. Discussion

This systematic review summarized the results of the effectiveness of the use of lower-cost equipment as an intervention instrument and/or to evaluate the use of feedback in changing the biomechanics of running. It was possible to identify that lower-cost equipment such as a metronomes, smartwatches, digital cameras, socks and insoles instrumented with sensors, and pressure platforms were used in the retraining of the cadence, foot strike pattern, and distribution of plantar pressure during running. We also observed that the majority of studies that performed running retraining interventions with feedback used an accelerometer to provide real-time tibial PPA information.

### 4.1. Effectiveness of Lower-Cost Equipment for Alterations in Spatio-Temporal, Kinetic, and Kinematic Variables of the Foot

Increasing cadence alone by using a metronome can be an effective way to change the foot strike pattern for some runners. In a study by Allen et al. [40], increasing cadence by 10–15% led to a significant change from heel strike to midfoot or forefoot stride by 17.5 and 30%, respectively. In a study by Miller et al. [58], the metronome was set at 180 steps/minute to provide real-time audio feedback during retraining, and the cadence significantly increased by 6.2%. All participants transitioned to a different type of foot strike than a heel strike after retraining. In the studies by Phanpho, Rao, and Moffat and Musgierd et al. [56,59], the metronome was used as an intervention instrument to control the increase in cadence in relation to the preferred cadence. These results provided strong evidence that demonstrated the effectiveness of the metronome for use in running retraining in order to increase cadence.

Technological advances have created the possibility of the undertaking of running training outdoors through the insertion of wearables such as smartwatches that allow real-time feedback of certain components of the run [70]. In this way, the smartwatch enables retraining interventions in a runner's normal running environment, thereby potentially increasing the likelihood that individuals will adhere to the new running pattern. In a study by Baumgartner et al. [55], a smartwatch with an accelerometer was used to control cadence, and the experimental group demonstrated an increase of 8.6% in cadence after a six-week intervention. In a study by Willy et al. [54], the experimental group showed the same significant increase in cadence after eight retraining sessions; after one month, the increase was 8.5% in relation to baseline. In a study by Musgierd et al. [59], participants received a smartwatch to monitor their cadence and running pace at baseline. In a second session, which was performed with a 10% increased cadence using feedback from the metronome, the smartwatch guided the maintenance of cadence in real time. In that study, there was a significant overall increase of 7.3% in cadence between sessions. These results demonstrated strong evidence for the use of some smartwatches in running retraining in order to increase cadence.

Digital cameras can be utilized to obtain video of a run in the sagittal plane to evaluate cadence and also to detect the foot strike pattern. In a study by Sellés-Pérez et al. [60], a smartphone device was used to obtain video of a run in the sagittal plane and to evaluate the cadence in a group that performed running retraining to increase cadence. This group was required to follow a music rhythm that was increased by 10% in relation to the baseline cadence. After six weeks of retraining, it was found that the group significantly increased their cadence by 7.3%. In a study by Morris et al. [48], in the group that performed the retraining to transition from a heel strike to a non-heel strike and run at a cadence of 180 steps/minute (increased cadence), 80% of runners changed from a heel strike to a non-heel strike pattern after the two-hour training session. There was a slight but non-significant decrease in the percentage of non-heel strike runners at the 6-month and 1-year follow-up. Runners also presented significant increases in cadence from the baseline to post-training (approximately 6%), from the baseline to 6 months (approximately 3.7%), and from the baseline to 1 year (approximately 4.2%) of follow-up. Therefore, it can be argued

that there is strong evidence that clinicians, coaches, and researchers can confidently use video analysis to guide their gait retraining interventions for runners.

Instrumented socks are a recently developed wearable technology that consist of a Bluetooth-enabled instrumented socks that can provide real-time biofeedback to runners [47]. When this device is paired with a smartphone, the user can receive real-time auditory and/or visual biofeedback on their foot strike pattern, cadence, running pace, total distance covered, and elevation changes. In a study by Goss et al. [47], instrumented socks were used to detect the foot strike pattern using plantar pressure sensors and, through a smartphone application, provide real-time audio biofeedback on the distance covered, pace, foot strike pattern, and cadence for the runners who performed the retraining. In that study, the authors required that the runners not step on their heels, attempt to lean forward to step on the forefoot, and maintained a cadence of 180 steps/minute. The findings showed that 95% of the runners transitioned to a type of non-heel strike and that the majority (89%) maintained this transition in foot strike pattern after one month. In a study by Phanpho, Rao, and Moffat [56], instrumented socks with sensors were also used in the intervention to alter the foot strike pattern through visual feedback, and insoles instrumented with pressure sensors were used to determine the location of the center of pressure on initial contact of the foot with the ground. Initially, with a 10% increase in cadence, the authors obtained a modest anterior displacement of the center of pressure in the initial contact and consistent changes in relation to a lesser inclination of the foot in the initial contact, which corroborated other studies that used a 10% increase in cadence [24,40,42]. However, when there was a change in the foot strike pattern combined with an increase in cadence, the anterior displacement of the center of pressure was significantly more pronounced, which suggested that the average location of the center of pressure along the longitudinal axis of the insole changed by 192.7% in relation to the baseline. According to Stoltenberg et al. [71], the sensor-instrumented socks demonstrated moderate reliability for the detection of the foot strike pattern and excellent reliability in determining cadence. In a study by Musgierd et al. [59] that used insoles with sensors to collect cadence and peak force, it was identified that a 7.3% increase in cadence generated a decrease in the peak force of 5.6%. These results provided moderate evidence that the use of instrumented socks and insoles in running retraining to increase cadence and a non-heel strike pattern can improve important biomechanical factors related to injury risk.

Pressure sensors were also used with a pressure platform in a study by da Silva Neto, Lopes, and Ribeiro [57], who evaluated the effects of a running retraining strategy with feedback on the distribution of plantar pressure and plantar arch in runners who adopted a heel pattern. In that study, it was found that the group that received eight retraining sessions with visual feedback presented a reduction in plantar pressure on the heels, and the plantar arch during running showed a better adjustment in plantar support. These results demonstrated limited evidence for the use of the pressure platform in running retraining.

Collectively, the effectiveness of metronomes, smartwatches, digital cameras, and instrumented socks and insoles, which are lower in cost and consequently clinically accessible, in retraining programs that aim at the intervention and/or evaluation of the results of the cadence and foot strike pattern can be effective in reducing the impact load variables.

*4.2. Effectiveness of Lower-Cost Equipment for Changes in the Peak Positive Acceleration of the Tibia and Footwear*

Several studies used kinetic measures to analyze impact loads (forces applied to the skeleton when contact with the ground occurs) because these measures demonstrate a potential association with injuries due to overuse of the lower limbs [72–76]. For the analysis of impact loads, the most commonly used variables are peak vertical impact, average vertical load rate, instantaneous vertical load rate, tibia and footwear PPA, shock attenuation, vertical stiffness, and leg/lower extremity stiffness [35].

Accelerometers, which are small, lightweight devices, make it possible to identify the PPA of the tibia, which is strongly correlated with mean vertical force and instantaneous

vertical load rates, two common biomechanical parameters that indicate impact load and are associated with running-related injuries [77]. Previous studies used laboratory equipment such as force platforms or instrumented treadmills to provide biofeedback of kinetic parameters to participants [78,79]. However, the accessibility of biomechanics laboratories for clinical applications is generally limited, and treadmills equipped with force transducers are expensive equipment and are not readily available. On the other hand, accelerometers are relatively inexpensive compared to instrumented treadmills and are readily available [61]. Therefore, lightweight accelerometers have been used as a substitute instrument for impact load estimation in the absence of instrumented treadmills [31].

In the current study, we found that accelerometers were used as a tool for intervention through the provision of real-time feedback in 91.7% of the analyzed studies (Tables 4 and 5); in 54.5% of the studies the feedback was visual [31,32,61,62,65,66], in 18.2% it was auditory [30,64], in 9.1% it was tactile [68], in 9.1% it was auditory and visual [63], and in 9.1% it was auditory and verbal [69]. Only the study by Letafatkar et al. [67] used an accelerometer to assess the results of the intervention in relation to the PPA of the tibia.

Based on the results of studies that performed interventions with the use of an accelerometer to reduce the PPA of the tibia and/or footwear, a significant mean decrease of ~31% in PPA values between pre- and post-training was identified; in some studies, these values were close to 50%. In a study by Crowell and Davis [62], a sample constituted by runners with a tibial PPA >8 g and an intervention of eight sessions was conducted in which the runners were required to run more smoothly, make the steps silently, and (through visualization of a screen) keep their PPA below a horizontal line that represented 50% of the average PPA in the pre-training. The results showed that the tibial PPA decreased 48% after retraining, which was maintained in the one-month follow-up. In a study by Sheerin et al. [68] that was conducted using runners with high tibial acceleration and an intervention of eight sessions, runners were required to run with softer steps and eliminate the tactile feedback of vibration, the threshold of which was 10% below the tibial acceleration that resulted from the baseline. The results showed that the tibial acceleration from pre- to post-intervention when running on a treadmill decreased by 50%; while for running on the ground, it decreased by 28%.

Based on the above, it can be stated that there is strong evidence of the effectiveness of accelerometers in interventions with feedback for running retraining. According to Crowell et al. [61], the main advantage of accelerometer feedback is that a therapist or coach is not obliged to observe each step and provide feedback. In addition, accelerometer feedback provides a quantitative indication of a runner's progress. If the retraining program relied solely on verbal feedback, the only quantitative assessment of the runner's performance would come from the post-training data collection. Therefore, accelerometry provides a feedback method with potential applications in a wide variety of environments such as clinics, university laboratories, gyms, and while training outdoors on different surfaces.

### 4.3. Limitations of Studies and Future Directions

Some limitations should be considered when interpreting the findings of this review. To quantitatively evaluate the articles, the AXIS tool was used, which showed that all studies presented a very high quality for a cross-sectional design. However, the majority of the studies were unable to justify the sample size or its representativeness as a target population or to explain how they recruited the runners. This information was important to ensure that the recreational runners evaluated in these studies truly represented the population of interest. There were also significant limitations for some of the retraining studies: 54% of the studies did not incorporate a control group (Tables 3 and 5), which is essential to determine the effectiveness of the intervention. Still, in relation to the sample, of the 22 studies included, only the studies by Goss et al. and Miller et al. [47,58] examined the effects of real-time feedback in runners who had presented injuries in the past 12 months, which limited the clinical applicability of the evidence-synthesis findings. Although the use of biofeedback in healthy runner populations demonstrates the clinical feasibility of

this intervention potentially for purpose or prevention, there is no evidence that achieving the biomechanical modifications sought during feedback training will positively affect pain and functional outcomes in an injured population [34].

In relation to the intervention protocols performed in the studies, a great variability in duration was found: 54% of the studies had eight or more interventions [30–32,47,54,55,57,60,62,66–68] and 27% of the studies performed the intervention through a single session to verify the immediate effects of feedback [48,56,61,63–65]. Variability was also noted in the instructions provided to the experimental group, and 68% of the studies instructed the sample to run more smoothly, more quietly, and/or with the PPA below a threshold, which ranged from 10% to 50% in relation to the baseline depending on the study [30–32,47,48,57,58,61,62,64–69]; 27% of the studies instructed the sample to increase the rate of steps [40,54–56,59,60]; and one study did not provide instructions to the participants [63]. In addition, only two studies [48,67] included an assessment of the long-term running retraining retention (12 months) that met the Cochrane Group guidelines [80]. As a result, it is unclear what effect some of these interventions might have once incorporated into a habitual running pattern rather than being tested when the intervention is still new.

We identified only nine (41%) studies [40,48,55–57,59,60,65,69] that included only lower-cost equipment in their methodology to conduct the intervention/assessment. As a result, the methodological replicability of many studies was limited to most clinicians and trainers due to the lack of accessibility to high-cost equipment. There was a trend in recent studies to use an accelerometer to examine the external impact load during running, but few studies used an accelerometer for monitoring outside the laboratory environment in order to make the intervention/assessment more ecological. It is necessary for future studies that use lower-cost equipment in their interventions to effectively report their validation. Finally, there is a need for future studies that include new wearable technologies that provide feedback, such as instrumented socks and insoles, in order to identify their validity in terms of monitoring biomechanical variables and their effectiveness in running retraining programs.

## 5. Conclusions

Strong evidence suggested that metronomes, smartwatches, and digital cameras are effective in running retraining programs for intervening in and/or evaluating the results of cadence and the foot strike pattern. There was also moderate evidence of the use of socks and insoles instrumented with sensors and limited evidence of the use of pressure platforms to analyze the distribution of plantar pressure and peak force in contact with the ground. Accelerometers, on the other hand, presented strong evidence of effectiveness in studies that performed interventions with feedback to reduce the PPA of the tibia and/or footwear during running. Finally, there was a lack of studies that used exclusively lower-cost equipment to perform the intervention in/assessment of running retraining, as well as those that used this equipment in more ecological environments and that analyzed the retention of running retraining in the long term.

**Author Contributions:** Conceptualization, L.M.D. and R.R.B.; methodology, L.M.D., J.R.C. and R.R.B.; validation L.M.D., V.C., F.A.M. and R.R.B.; formal analysis L.M.D., V.C., J.R.C., F.A.M. and R.R.B.; investigation, L.M.D., V.C., and R.R.B.; writing—original draft preparation, L.M.D.; writing—review and editing, L.M.D., J.R.C., F.A.M. and R.R.B.; supervision, F.A.M. and R.R.B. All authors have read and agreed to the published version of the manuscript.

**Funding:** Authors acknowledge the financial support from the Coordenação de Aperfeiçoamento de Pessoal de Nível Superior, Brazil (Finance Code: 001).

**Data Availability Statement:** Publicly available datasets were analyzed in this study. This data can be found here: https://osf.io/z4uxm.

**Conflicts of Interest:** The authors declare no conflict of interest.

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
