# Peer review of "Effectiveness of Lower-Cost Strategies for Running Gait Retraining: A Systematic Review"

_applsci, doi:10.3390/app13031376_

Round 1

Reviewer 1 Report

The paper describes a systematic literature review on the effectiveness of low-cost strategies for running gait retraining. The authors have presented the literature review very meaningfully and included major studies. I have the following comments on the paper.

Abstract: 

·         Line 22: What are possible gait retraining interventions? Is it only step cadence and foot strike patterns? The abstract needs rewriting to reflect what is presented in the paper.

Introduction: 

·         Introduction of Problem: A low-cost strategy may not accurately measure the biomechanical outcomes related to the injuries. Knowing a list of biomechanical measures required to assess the retraining strategy is essential. What are possible gait retraining interventions?

·         Line 38: as described, alterations in running mechanics are many to avoid running injuries. It is important to clarify why the authors have chosen only two types of alterations, as given in the abstract.

·         Briefly describe the background by illustrating the general area of research. It also mentions the importance of the selected research area by highlighting its critical factors.

·         Line 46: How this literature review differs from the systematic review conducted in [20]? It is better to describe relevant surveys on this topic and describe the features that are not covered in those surveys. It will hence validate the importance of the literature review done by authors.

·         Line 58: Please link the variables “step cadence, foot strike angle, and foot strike pattern” with the outcome measures required in the retraining interventions.

·         Line 62: I think verification of the low-cost sensors is beyond the scope of this paper. Also, I don’t see any proof of this statement.

 Background and Related Work: 

·         Classify the existing methods into some categories.

·         It is important to know whether low-cost equipment accurately measures the biomechanical measures required to assess gait retraining.

Selection of Studies

·         Line 91: One exclusion criterion is “used expensive equipment for analysis” whereas in the tables, both low-cost and high-cost equipment is mentioned. So, this exclusion criterion must be modified.

Results

·         Table 2, row 2, Is pressure platform considered as low-cost equipment?

·         All tables: Column 4 lists the equipment used, column 5 describes the variables analyzed from low-cost equipment only, and similarly, column 6 shows the result (low-cost equipment). How can we separate the variables and results depending on low-cost and high-cost equipment? Are the results mutually exclusive in these two categories?

Discussion

·         Line 232: How smartwatch can guide the maintenance of running pace without feedback? If there is visual feedback from the smartwatch, is it real-time or after-session feedback?

·         Line 236 onwards: It will be interesting for the readers to know the relationship between increased cadence and improved heel strike patterns. What is the reason for such an improvement?

Round 2

Reviewer 1 Report

The authors have answered all of my comments and revised the paper accordingly.

Author Response

We thank the reviewer for this comment.

Reviewer 2 Report

Thank you to the authors for their time spent revising the manuscript. In the future, I suggest you add your in-text changes to the response to reviewer document. Saying “We adjusted this information in the text” gives the reviewer no indication of what specific changes you made to address our comments.

My remaining comments are minor.

Line 60-63: You seem to selectively not cite studies that found no change in injury risk when impact loads were reduced. Please provide these citations and comment on these results.

Line 68-70: Please clarifying that there is converging evidence that reducing running impacts is NOT associated with a reduction in injury risk and cite accordingly.

Line 86-87: Same comment as above

To address my previous comment: “Line 169 (now line 23): What does “clinical success” mean. If I am interpreting correctly, the device was successful at modifying preferred cadence, but it is unknown if modifying this biomechanical outcome is clinically relevant for reducing injury risk?” You just changed the word from “clinical success” to “effectiveness”. What does “effective” mean? Please clarify it was effective at modifying step frequency, but it is unknown if it was successful at modifying injury risk.
